# HarnessLLM: Automatic Testing Harness Generation via Reinforcement Learning

## Abstract

Existing LLM-based automatic test generation methods mainly produce input and expected output pairs to categorize the intended behavior of correct programs. Although straightforward, these methods have limited diversity in generated tests and cannot provide enough debugging information. We propose HarnessLLM, a two-stage training pipeline that enables LLMs to write harness code for testing. Particularly, LLMs generate code that synthesizes inputs and validates the observed outputs, allowing complex test cases and flexible output validation such as invariant checking. To achieve this, we train LLMs with SFT followed by RLVR with a customized reward design. Experiments show that HarnessLLM outperforms input-output-based testing in bug finding and testing strategy diversity. HarnessLLM further benefits the code generation performance through test-time scaling with our generated test cases as inference-phase validation.

## 1 Introduction

Large language models (LLMs) have demonstrated remarkable proficiency in code-related tasks, including code generation, completion, and even resolving software engineering issues through tool use Chen et al. (2021); Li et al. (2022); OpenAI (2024); DeepSeek-AI (2025); Jimenez et al. (2024); Li et al. (2025). However, compared to these code generation tasks, automatic testing and debugging AI-generated programs have received comparatively little attention, even though comprehensive test suites are critical for ensuring the correctness and robustness of the AI-generated code Chen et al. (2024a); Prasad et al. (2025); Sinha et al. (2025); He et al. (2025b); Zhang et al. (2023a).

Existing works in automatic testing mainly prompt the language model to generate input-output pairs that characterize the intended behavior of correct programs Chen et al. (2022); Prasad et al. (2025); Zeng et al. (2025); Lin et al. (2025). As depicted in Figure 1, the model produces examples of test inputs alongside their expected test outputs. The target program is then executed on a test input, and the output that the program generates is compared against the corresponding expected test output. A bug is exposed if the two outputs diverge.

Although straightforward, such an input-output test case generation paradigm has two potential drawbacks. *First*, the test inputs generated by the language model tend to be simple and homogeneous, so they may not have sufficient coverage of the sophisticated corner cases that could expose bugs. *Second*, such a paradigm requires that the model generates the correct output by itself, which becomes extremely

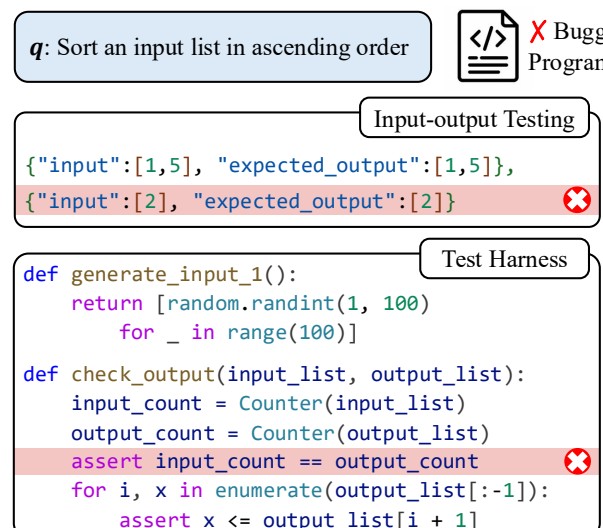

Figure 1: Comparison between input-output pairs (top) and test harness (bottom).

challenging for complicated programming tasks or for complicated test inputs. In short, the fundamental paradox is that the 'tester', *i.e.,* the language model that generates test cases, is often much weaker than the 'testee', *i.e.,* the program, in accomplishing the complex programming tasks (otherwise, the language models would not have to rely on code generation to solve these tasks).

In this paper, we explore a novel debugging paradigm that could resolve this paradox – LLM-based *test harness generation*. Rather than letting the language model directly generate input-output pairs, we prompt it to write executable code, a matching rival to the testee, to generate test inputs and validate target program outputs. In this way, both aforementioned drawbacks can be addressed simultaneously. On the input side, executable code can easily generate various richly structured, diverse, and complicated inputs. On the output side, executable code opens up many possibilities to validate target program outputs. It can ❶ directly generate hardcoded expected output, as does the input-output paradigm, or ❷ write a reference program to compute the expected output, or, most interestingly ❸ assert output properties and requirements. As shown in Figure 1, for a program that sorts a list of input integers, the LLM first writes an input generator, `generate_input_1`, to generate random lists, which are fed to the target program for execution. The returned outputs are then validated by an LLM-defined function, `check_output`, which checks that the result is sorted and preserves the original integers. With programmatic input generation and output validation, testing harnesses can support complex invariant checking and stress testing, enabling more comprehensive testing and detection of deep logical bugs.

However, the key challenge of the test harness generation paradigm is that even the strong coding LLMs are not inherently capable of test code generation, which requires different skills than generating code for programming tasks: The former requires understanding the given program's logic, control and data flow, designing proper stress tests, and writing validation logic, while the latter is mainly about writing code to fulfill the required functionality. To demonstrate this, our initial experiment compares the bug-finding rates of the input–output strategy versus test harness generation on the LiveCodeBench and Codeforces datasets Jain et al. (2025); Penedo et al. (2025), using a strong reasoning model `Qwen3-32B` Yang et al. (2025a). Surprisingly, direct prompting for test harnesses does not yield better bug finding capabilities (details in Table 1).

To close this gap, we propose HarnessLLM, a two-stage training pipeline combining supervised fine-tuning (SFT) with reinforcement learning (RL) with customized reward functions. First, we collect SFT data by prompting `Qwen3-32B` and filtering for harnesses that successfully expose a bug. We warm up a smaller model (*e.g.,* `Qwen3-4B`) with SFT on collected data. The purpose of this stage is to provide a reasonable starting point for reinforcement learning, which improves RL's training efficiency. Second, we further train the SFT model using RL with our customized verifiable outcome reward. Here, we assume access to a ground-truth program during training. To encourage the model to generate valid harnesses, we first give a zero reward to generated tests that trigger compilation or runtime errors on the ground-truth program. Then, we design rewards to incentivize the model to generate effective tests that crash the target programs. Specifically, a positive reward is assigned when the ground-truth program can pass the generated tests but the target program fails, indicating that the test harness correctly identifies bugs in the target program. We train the model to maximize the expected reward using the GRPO algorithm Shao et al. (2024). The RL training can further strengthen the model's capabilities to generate effective test harnesses and improve the model's generalizability.

We train on two base models (`Qwen3-4B` and `Llama3.2-3B` Llama (2024)) and evaluate on four benchmarks containing buggy programs. Experiments show that our model outperforms all baselines, including the off-the-shelf `Qwen3-32B` and another model that is also trained with RL but only generates input-output pairs. Moreover, the learned harness generator generalizes to code produced by unseen models and can be used for improving code generation performance. Specifically, using the execution results of generated test cases to select the best out of 8 responses improves `Qwen3-32B`'s performance from 63.5% to 69.5% on LiveCodeBench. To the best of our knowledge, HarnessLLM is *the first LLM-based testing harness generation that enables comprehensive testing and benefits competitive programming tasks.*

## 2 Related Works

**Automatic Test Case Synthesis.** Test cases are crucial in evaluating code correctness. While many established benchmarks rely on manually written test cases Chen et al. (2021); Austin et al. (2021); Hendrycks et al. (2021), this process is labor-intensive and does not scale well. To address this limitation, a variety of automatic test case synthesis methods have been proposed. Traditional approaches leverage programming language techniques to explore the input space and cover diverse execution paths Puspitasari et al. (2023); Forgács & Kovács (2024); Guo et al. (2024); Reid (1997). Although these techniques improve input coverage, they often fall short in capturing code semantic relationships and complex control flows, which can lead to undetected failures during runtime. Recently, LLMs have been used to synthesize test cases by directly generating inputs and expected outputs Yuan et al. (2024); Chen et al. (2024b); Han et al. (2024); Li & Yuan (2024); Guzu et al. (2025); Xiong et al. (2023); Wang et al. (2025a); Cao et al. (2025); Wang et al. (2025b); designing property-based tests to validate program behaviors Vikram et al. (2024); He et al. (2025a); Bose (2025); or integrating with software engineering techniques like fuzzing Xia et al. (2024); Lemieux et al. (2023); Yang et al. (2025b); Altmayer Pizzorno & Berger (2025). However, most of these works focus on designing a debugging pipeline with frozen LLMs, whereas our goal is to improve LLMs' core ability to generate test cases through SFT and RL training. Additionally, we introduce a novel paradigm that shifts from direct output prediction to execution-based output validation. Our HarnessLLM programmatically generates inputs and validates outputs, expanding the design space of test cases.

**Reinforcement Learning with Verifiable Rewards.** Reinforcement learning has shown great potential in improving LLM abilities in many domains requiring heavy reasoning, such as math problem solving DeepSeek-AI (2025); Kimi (2025); Shao et al. (2024); Yu et al. (2025); Hou et al. (2025), code generation Le et al. (2022); El-Kishky et al. (2025); Liu & Zhang (2025), and robotic control Chu et al. (2023); Ji et al. (2025). In this work, we use RL to improve LLMs' test case generation capabilities. By designing a customized reward that judges whether the generated test cases can differentiate between correct and buggy programs, we train LLMs to learn the reasoning skills required to write effective test cases.

## 3 Methodology

### 3.1 Problem Formulation

Formally, let $q$ be the description of a programming problem with input space $\mathcal{I}$ and output space $\mathcal{O}$. Denote $f, g : \mathcal{I} \to \mathcal{O}$ as two programs for this problem, where $f$ is a potentially buggy implementation that is under testing, and $g$ is a ground-truth implementation for the problem. We say $f$ has logical bugs if for some $\boldsymbol{x} \in \mathcal{I}, f(\boldsymbol{x}) \neq g(\boldsymbol{x})$. In other words, $\boldsymbol{x}$ triggers the divergent behaviors of the buggy and reference programs. Therefore, an automatic debugging method generally contains two steps: generating inputs that can potentially trigger the bug and comparing the target program's output with the reference output.

However, in most real-world situations, the ground-truth implementation $g$ is not available, which necessitates an approximate verifier to validate the output of $f$. Denote this verifier as $v : \mathcal{I} \times \mathcal{O} \to \{0, 1\}$, where $v(\boldsymbol{x}, \boldsymbol{y}) = 1$ indicates that output $\boldsymbol{y}$ on input $\boldsymbol{x}$ is deemed correct. Our goal in this paper is to train an LLM for automatic debugging that, given $q$ and $f$, emits both a set of inputs $\{\boldsymbol{x}_i\}_{i=1}^N$ and a corresponding verifier $v$. Note that we mainly focus on finding *logical bugs* in a target program, *i.e.,* deviations from intended behavior, and leave security vulnerability for future work.

**Challenge of Input-Output Testing.** The input-output testing can be considered as having a simple verifier that compares the program's output with the expected output. Specifically, the model generates a set of pairs $\{(\boldsymbol{x}_i, \hat{\boldsymbol{y}}_i)\}_{i=1}^N$, where $\hat{\boldsymbol{y}}_i$ is the expected output for input $\boldsymbol{x}_i$. The verifier is then an indicator function $v(\boldsymbol{x}_i, f(\boldsymbol{x}_i)) = \mathbb{1}(f(\boldsymbol{x}_i) = \hat{\boldsymbol{y}}_i)$. However, this simple verifier requires the model itself to come up with a correct expected output, which limits the complexity of test cases. In the following, we propose a framework that generates test harnesses to address this challenge.

### 3.2 Generating Test Harness for Debugging

We propose instead that the LLM writes a test harness code that synthesizes inputs and programmatically checks outputs. Having harnesses can help produce more diverse testing cases and provide more valuable feedback when the program crashes. Specifically, our framework consists of three steps.

***Step 1: Generate Input.*** The model implements a set of input generators, each returning a list of inputs for the program (*e.g.,* `generate_input_1()`). By leveraging loops or random functions, the LLM can craft rich test inputs, which would be difficult to get with hardcoding.

***Step 2: Execute.*** Each generated input is fed to the program $f$, and the resulting output is captured.

***Step 3: Validate Output.*** A model-implemented function `check_output(input,output)` is used to validate the correctness of each captured output. The model can use various ways for validation, such as checking specific invariants or comparing with output from a brute-force implementation. This output checker uses assertions to check correctness, and a bug is reported if the assertions fail for *any* pair of generated input and captured output.

Figure 7 shows a complete example of model generation for this process, and Figure 9 shows the detailed prompt we use.

### 3.3 Improving Test Harness via RLVR

Despite the promise, we found off-the-shelf LLMs struggle to generate effective harnesses. To remedy this, we design a two-stage training pipeline to improve their performance. Figure 2 depicts an overview of our pipeline.

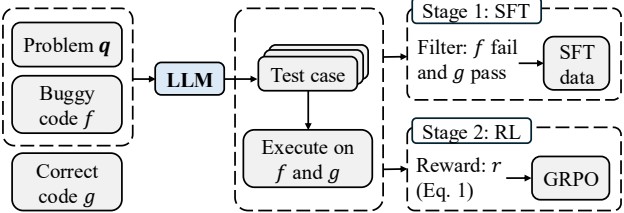

Figure 2: Overview of our training pipeline.

**Stage 1: SFT Warm-Up.** We prompt `Qwen3-32B` to generate test harnesses as described in Section 3.2. The model response contains a long reasoning chain and a final code block. We execute the harnesses against both the target program $f$ and the ground-truth program $g$ and retain only responses for which $g$ passes but $f$ fails. We then fine-tune a smaller model (*e.g.,* `Qwen3-4B`) with SFT on the filtered dataset. The SFT model has a basic understanding and skills for test harness generation. Using it as an initialization for RL can improve the learning efficiency of RL, as the early training stage can receive some meaningful positive rewards.

**Stage 2: RL with Verifiable Outcome Reward.** To further improve the generalizability of the warmed-up model, we follow recent works to train the model with RL against a verifiable outcome reward DeepSeek-AI (2025); Lambert et al. (2025). Specifically, for each rollout $o$ the model generates, let $\{\boldsymbol{x}_i\}_{i=1}^N$ be the corresponding inputs, we define the following reward function based on the execution results on $f$ and $g$:

$$r(o; f, g) = \begin{cases} 1, & \text{if } g \text{ passes and } f \text{ fails}; \\ 0.1, & \text{if } g \text{ fails}^1 \text{or } f \text{ passes,and } \exists\, \boldsymbol{x}_i : f(\boldsymbol{x}_i) \neq g(\boldsymbol{x}_i); \\ 0, & \text{otherwise} \end{cases} \tag{1}$$

In other words, a reward of 1 is given only when the ground-truth program can pass the test, but not the buggy program, indicating a correct test case. Otherwise, if all inputs are valid (*i.e.,* they do not trigger runtime errors on $g$) and at least one input can trigger different outputs for $f$ and $g$, we assign a partial reward of 0.1, which encourages the model to generate bug-exposing inputs. Note that in this case, the input generators work well, but the output verifier generates ineffective assertions, which either fail the correct code $g$ or do not crash the buggy code $f$. Nevertheless, we still assign a partial reward to incentivize the model to generate good inputs. Finally, a reward of 0 is given when no input can expose the bug. Importantly, the requirement that $g$ has to pass the generated test cases prevents the model from hacking rewards by generating arbitrary invalid tests. We maximize the expected reward using GRPO.

---

[1]Assertion errors in output verifier. All inputs still need to be valid, *i.e.,* do not trigger runtime errors on $g$.

### 3.4 Data Collection

Both training stages in Section 3.3 require data in the format of a problem description $q$, a buggy implementation $f$, and a ground-truth implementation $g$. To collect such data, we follow prior works Luo et al. (2025) to source from existing datasets of coding problems, including TACO Li et al. (2023), SYNTHETIC-1 Intellect (2025), LeetCode Xia et al. (2025), and Codeforces MatrixStudio (2025). The original solution in the datasets is used as ground-truth program $g$, after an additional round of filtering to make sure $g$ passes all provided test cases of the problem.

To collect the buggy programs $f$, we prompt a series of LLMs to solve the problem, including `Qwen2.5-Coder` 1.5-7B Hui et al. (2024) and `DeepSeek-R1-Distill-Qwen-1.5B` DeepSeek-AI (2025). We only keep programs that satisfy both of the following conditions: ① The program passes the demo test cases in the problem description; and ② The program fails on at least one test case of the problem. This makes sure the retained programs are partially correct but still have bugs. We retain at most two buggy programs per problem and select the two that pass the most test cases if multiple programs satisfy the two conditions.

After decontamination against all evaluation data, the training set contains 12,043 unique $(q, f, g)$ triplets. We use all samples for RL training and a subset of 6,805 samples for SFT. Appendix A.1 details our data collection procedure.

## 4 Experiments

We conduct experiments to validate the effectiveness of HarnessLLM. Specifically, we aim to answer two questions: ① Does our two-stage training pipeline enhance models' ability to write test harnesses? ② Does harness-based testing outperform input-output testing in identifying bugs?

### 4.1 Experiment Setting

**Evaluation Benchmarks.** We mainly evaluate on three widely used code generation datasets: MBPP+ Austin et al. (2021); Liu et al. (2023), LIVECODEBENCH Jain et al. (2025), and CODEFORCES Penedo et al. (2025). We repurpose these datasets for the bug detection task by collecting triplets of problem description, buggy program, and ground-truth program.[2] For MBPP+, we directly use the split MBPP+FIX (HARD) in `UTGen-32B` Prasad et al. (2025). For LIVECODEBENCH and CODEFORCES, we follow the procedure described in Section 3.4. Particularly, we create two dataset variants: ❶ SEEN version contains buggy programs generated by `DeepSeek-R1-Distill-Qwen-1.5B`, which is also used to generate our training data. ❷ UNSEEN version contains buggy programs generated by `Qwen3-14B`, which is never seen during training, and evaluates the generalizability of our models to different code generators (details in Appendix A.2). Additionally, we also test on ULT Huang et al. (2025), which targets functions for real-world coding problems beyond competitive programming.

**Metrics.** We extend the three standard metrics proposed in Prasad et al. (2025) for test harnesses. Specifically, ❶ **Good input (GI)** calculates the percentage of responses that have at least one bug-exposing input, *i.e.,* $\exists \boldsymbol{x}_i : f(\boldsymbol{x}_i) \neq g(\boldsymbol{x}_i)$. This metric purely measures the ability of the input generator. ❷ **Invalid test rate (ITR)** measures the percentage of responses where the ground-truth program fails, *e.g.,* tests that have invalid inputs or incorrect assertions. ❸ **True bug rate (TBR)** measures the percentage of responses that correctly expose the bug, *i.e.,* the ground-truth program passes the tests but the buggy program fails. This metric assesses the *overall performance.*

For each input pair of problem and buggy program, we sample 8 responses and report the average performance of these 8 runs. We follow the official setting of Qwen3 to set the temperature at 0.6 and add a presence penalty of 1.5 Yang et al. (2025a). The maximum generation length is set at 32,000.

**Number of Test Cases.** We allow each model response to contain one or more test cases. For input-output testing, each model response could contain multiple pairs of input and expected output. For test harnesses, each response could contain multiple input generators, and each generator could generate multiple test

---

[2]All data are decontaminated by problem description.

Table 1: Performance on finding bugs (average of 8 runs). *: The model and training set are not released, so we compare with the number reported in the original paper. Note that the results of `Qwen3-32B` come from the original model without any fine-tuning.

| | MBPP+Fix (Hard) | | | LiveCodeBench | | | Codeforces | | |
|---|---|---|---|---|---|---|---|---|---|
| | GI ↑ | ITR ↓ | TBR ↑ | GI ↑ | ITR ↓ | TBR ↑ | GI ↑ | ITR ↓ | TBR ↑ |
| `UTGen-32B`* Prasad et al. (2025) | 56.1 | 40.8 | 34.7 | – | – | – | – | – | – |
| `Qwen3-32B (Input/Output)` | 56.4 | 10.1 | 49.3 | 56.7 | 5.1 | 54.8 | 79.9 | 21.6 | 67.1 |
| `Qwen3-32B (Harness)` | 78.7 | 11.9 | 68.6 | 69.1 | 15.5 | 55.1 | 80.4 | 33.9 | 54.8 |
| `Qwen3-4B` | | | | | | | | | |
| `SFT (Input/Output)` | 52.1 | 11.3 | 44.6 | 45.7 | 8.2 | 42.9 | 75.1 | 23.6 | 59.5 |
| `SFT (Harness)` | 78.1 | 17.7 | 62.9 | 60.4 | 23.7 | 42.1 | 82.5 | 46.9 | 46.0 |
| `RL (Input/Output)` | 82.5 | 13.9 | 72.7 | 68.4 | 9.9 | 65.1 | 89.8 | 21.0 | 72.2 |
| `RL (Harness)` | **84.4** | **13.0** | **74.1** | **79.1** | **9.5** | **69.9** | **91.8** | **19.1** | **74.4** |

inputs. In preliminary experiments, we observe that the number of test cases in each response significantly affects the performance (details in Appendix B.1). Thus, for the teacher model and SFT models, we report the performance of the best number of test cases. Namely, 1 test case per response for input-output testing and 5 for test harnesses.[3] However, restricting the same number of test cases for all problems may be suboptimal. Therefore, during RL training, we allow the model to generate any number of test cases from 1 to 20, and the model learns the optimal number of test cases for each problem through training. The following section reports the performance of this setting. In Appendix C.7, we further show results when controlling the number of test cases.

**Baselines.** We mainly compare with the baseline that generates input-output pairs for testing. For fair comparison, we conduct the same two-stage training as our method. Particularly, we use the same teacher model to generate an equal amount of SFT data, and we use the same reward in Eq. 1 for RL training. Additionally, we report the performance of directly prompting `Qwen3-32B` with both testing strategies, as well as `UTGen-32B` Prasad et al. (2025), which also generates input-output pairs but is trained with only SFT without RL. Finally, we compare with `TrickCatcher` Liu et al. (2025), which is a complex LLM-powered, training-free pipeline for test case generation.

**Implementation Details.** We test our framework on `Qwen3-4B` and `Llama3.2-3B`. For SFT, we train all models for 15 epochs and select the best checkpoint based on the validation performance. For RL, we use the Verl framework Sheng et al. (2024) and train all models for 500 steps (detailed hyperparameters in Appendix B.2). We parallelize the reward calculation across all CPU cores, and on average, it takes 0.06 seconds to execute the test harnesses for each rollout during training. Appendix C.1 shows the RL training dynamics.

## 4.2  Main Results

**Ability to Find Bugs.** Table 1 shows the performance of `Qwen3-4B` on finding bugs generated by models that have been seen during training. There are two observations. **First**, our RL-trained model for test harness generation consistently outperforms the counterpart that generates input-output pairs. Specifically, it achieves better performance on all metrics across all benchmarks, demonstrating the benefits of test harness generation for both input generation and output verification. **Second**, both RL-trained small models surpass the 32B teacher models, which illustrates the effectiveness of our proposed two-stage training. Interestingly, although test harnesses initially underperform input-output generation on the teacher model and SFT models, our RL training unlocks their advantage and leads to better final performance.

Appendix C.2 shows results on `Llama3.2-3B`, which suggest that our method has better generalizability than input-output testing. Appendix C.3 presents the results on ULT, which demonstrate that our method has

---

[3]If a response contains more test cases, we only evaluate the first 1 or 5 test cases.

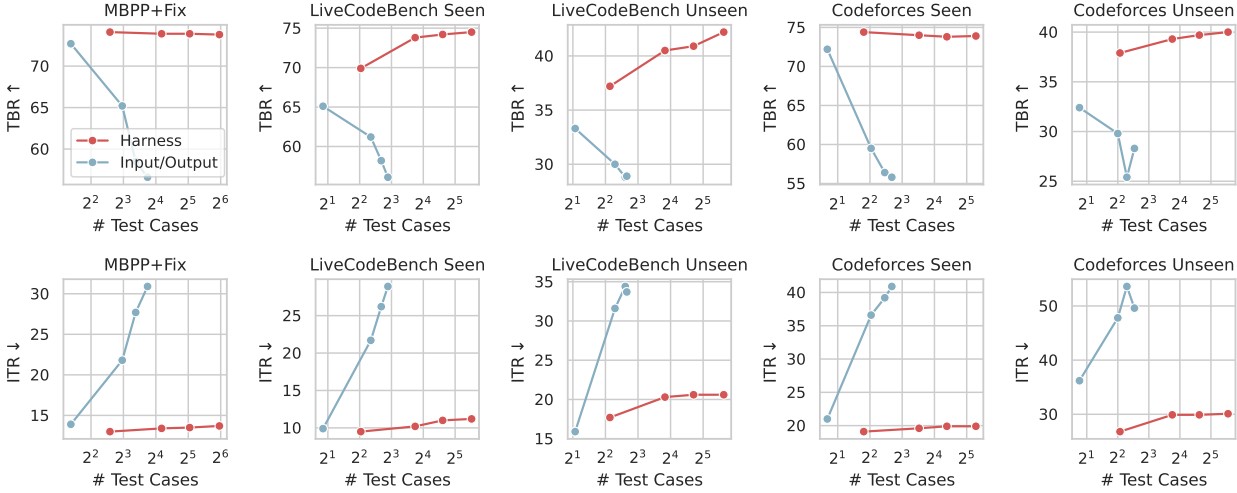

Figure 3: True bug rate (TBR) and invalid test rate (ITR) as the number of test cases increases.

higher test coverage than input-output testing. Finally, Appendix C.4 compares with `TrickCatcher`, which shows that our method is more efficient and achieves better performance.

**Generalizability to Unseen Models.** We next evaluate our models' ability to debug for models that have never been seen during training. Specifically, we collect buggy programs generated by `Qwen3-14B`. These buggy programs are different from those in Table 1 in two ways: ① They are from an unseen model and thus may have different distributions for the bugs in the code. ② They are from a stronger model and pass more test cases, so they contain deeper logical bugs. Performance shown in Table 2 illustrates similar observations as Table 1. Particularly, our RL-trained test harness generator substantially outperforms the model that generates input-output pairs. Appendix C.5 re-

Table 2: Generalization to unseen models. The buggy code is sampled from `Qwen3-14B`, which is not seen during training.

|  | **LiveCodeBench** | | | **Codeforces** | | |
|---|---|---|---|---|---|---|
|  | GI ↑ | ITR ↓ | TBR ↑ | GI ↑ | ITR ↓ | TBR ↑ |
| **Qwen3-32B** | | | | | | |
| I/O | 25.0 | 8.3 | 23.0 | 43.2 | 20.1 | 31.5 |
| Harness | 36.8 | 20.4 | 22.4 | 61.6 | 36.3 | 32.3 |
| **Qwen3-4B** | | | | | | |
| SFT (I/O) | 19.4 | 12.4 | 17.3 | 35.7 | 25.4 | 23.1 |
| SFT (Har) | 34.1 | 34.5 | 16.3 | 59.1 | 45.2 | 25.3 |
| RL (I/O) | 37.0 | **15.9** | 33.3 | 53.0 | 36.2 | 32.4 |
| RL (Har) | **51.1** | 17.7 | **37.2** | **67.3** | **26.8** | **37.9** |

peats the same experiment with buggy programs generated by `gpt-oss` OpenAI (2025), which shows similar trends as Table 2. These results show that our models can generalize to unseen models. It also validates that the improvements of our method are not overfitting to a particular distribution of bugs.

**Scaling Number of Test Cases.** In the experiments above, we have limited each response to at most 20 test cases. We next investigate if we can further improve the performance by increasing the number of test cases in each response. Specifically, we employ different strategies to scale up the number of test cases for baselines and our method. For the baseline that generates input-output pairs, we directly change the instruction to the LLM to ask it to generate more test cases. For our method, since many input generators use random functions to generate inputs, we simply run the input generators multiple times with different random seeds to get more test inputs. Figure 3 shows the performance of the RL-trained models with respect to the number of test cases. As can be observed, when generating more test cases for the baseline method, the percentage of correctly identified bugs (TBR) drops significantly, and the amount of invalid tests (ITR) quickly increases, leading to a much worse performance. The observation confirms the limitations of hardcoded input-output pairs, since the probability of getting all test cases correct decays exponentially when the number of test cases increases. On the contrary, for our method, TBR consistently increases

for three datasets and maintains the original value for the other two datasets, and ITR also demonstrates only a marginal increase. The results highlight two benefits of programmatic input generation and output verification: ❶ The input generator can *easily generate more inputs to increase the test coverage*; and ❷ The same output verifier can be reused for different inputs *without sacrificing the accuracy.*

**Using Feedback for Test-time Scaling.** Given the superior bug-finding performance of our model, we now explore its application to improve code generation via test-time scaling. Specifically, given a coding problem, we sample 8 candidate programs from an LLM and use the test case generator to generate test cases for each program. We collect all generated test cases for the same problem and run them against each candidate program. The program that passes the most test cases is selected as the final program.

Table 3: Best-of-8 performance on LIVECODEBENCH where the code is selected based on the execution results of the generated test cases.

| | Code Generator | | |
| --- | --- | --- | --- |
| | Qwen3-4B | Qwen3-14B | Qwen3-32B |
| Original pass@1 | 52.60 | 60.23 | 63.53 |
| RL (I/O) | 60.12 | 65.40 | 67.45 |
| RL (Harness) | **60.70** | **66.57** | **69.50** |

Table 3 shows the results on 341 problems of LIVECODEBENCH with three code generators. As can be observed, scaling with both test case generators significantly improves the performance of the original LLM (original pass@1). Furthermore, our model with test harnesses outperforms the input-output testing, demonstrating its superior performance in judging code correctness. The results also confirm that our model's improvements on finding bugs can be **translated into improved code generation**.

**Generalization to Other Programming Languages.** So far, we have mainly tested programs written in Python. To evaluate our model's generalizability to more diverse programming languages, we additionally evaluate on TrickyBugs Liu et al. (2024), which contains 250 human-written buggy programs in C++. We directly use the RL-trained models in Table 1 to generate test cases for these buggy programs. Results in Table 4 show that our model outperforms the input-output testing baseline, suggesting that our training improves the general capability to generate test harnesses for **diverse programming languages**.

Table 4: Generalization to buggy programs in C++.

| | TrickyBugs | | |
| --- | --- | --- | --- |
| | GI ↑ | ITR ↓ | TBR ↑ |
| RL (I/O) | 28.4 | 14.6 | 26.6 |
| RL (Har) | 33.0 | 16.1 | **27.1** |

### 4.3 Additional Analyses

**Performance across Difficulty Levels.** Section 4.2 reports aggregated performance across all problems in a dataset. We next investigate if the improvement of our method is consistent across problems with different difficulty levels. Figure 4 shows the detailed performance breakdown of the baseline and our method. Specifically, on LIVECODEBENCH, we use the original difficulty categories. On CODE-FORCES, we split problems based on their ratings (HARD corresponds to problems with ratings greater than 2400 and MEDIUM corresponds to problems with ratings greater than 1800). As can be observed, while the performance of both methods degrades when problems become harder, our method better maintains the performance compared to the baseline. The results indicate that test harnesses can better generalize to difficult problems, verifying our motivation that input-output testing is limited for complex problems.

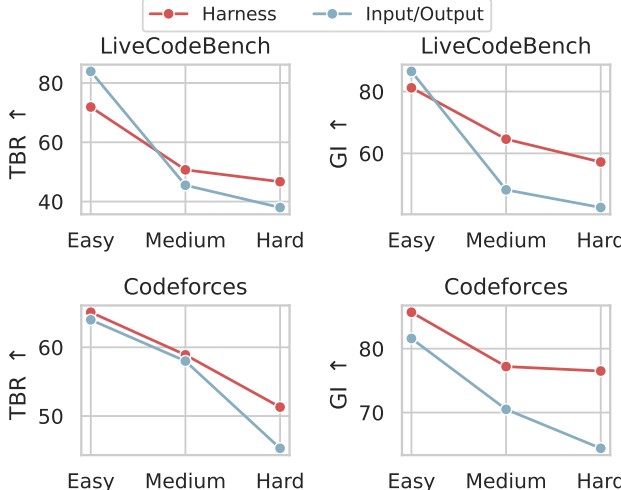

Figure 4: Performance across difficulty levels.

**Distribution of Testing Strategies.** By programmatically generating inputs and validating outputs, test harnesses allow models to have broader strategies for debugging. For example, we identify two main ways models use to generate inputs, which are explicitly emphasized in our SFT data: ① *Hardcoded*: models return a list of hardcoded inputs. ② *Dynamic*: models dynamically generates inputs with code (*e.g.,* randomized inputs through random functions). Similarly, we identify three ways models employ to validate a captured output: ① *Hardcoded*: models compare the output with a hardcoded expected output. ② *Compare reference*: models implement a reference solution (*e.g.,* a brute-force solution) and compare the output with that obtained from the reference solution. ③ *Check invariant*: models check if the output satisfies specific invariants such as the length and range.

We prompt `Qwen3-32B` to classify the strategies used in each response (details in Appendix B.3). Figure 5 shows the distributions of the input generator and the output verifier respectively. Specifically, we report input generator strategies for buggy programs that are mostly wrong (pass less than 25% of test cases), medium (pass 25% to 75% of test cases), and mostly correct (pass greater than 75% of test cases). As can be observed, when the buggy program is mostly wrong and has obvious bugs, the model generates more hardcoded inputs. When the buggy program is more correct and contains bugs that are hard to identify, the model generates more dynamic inputs to increase test coverage.

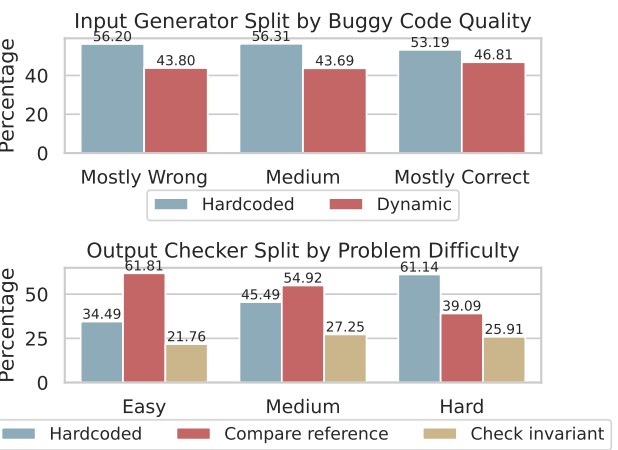

Figure 5: Distribution of testing strategies.

Similarly, when the problem is easy, the model more often implements a reference solution for validation;[4] and when the problem becomes difficult, the model hardcodes more expected outputs. The observations demonstrate that the model can adapt its testing strategies to specific problems. Figure 7 shows an example where the model combines multiple strategies for output validation.

**Diversity of Test Cases.** By programmatically generating inputs, our model can potentially generate diverse inputs that would be difficult to synthesize with hardcoding. We verify this by comparing the diversity of inputs generated by the baseline and our models. Specifically, we analyze the test cases generated for the programs of `Qwen3-32B` in Table 3. We evaluate a subset of 214 problems that take `stdin` as inputs. For fair comparison, we randomly downsample the generated test cases so that the two models have the same number of test cases. Results in Table 5 show that our method generates more diverse inputs and inputs with various lengths than the baseline (please refer to Appendix C.6 for details of the metrics).

Table 5: Diversity of inputs for test cases generated by the two models.

|  | Unique ratio ↑ | Length range ↑ | Length std ↑ |
|---|---|---|---|
| RL (I/O) | 48.6 | 1.00 | 0.31 |
| RL (Har) | 77.1 | 8.90 | 2.69 |

## 5 Conclusion

We propose HarnessLLM, a pipeline for training LLMs for test harness generation. Experiments demonstrate that HarnessLLM outperforms its counterpart that generates input-output pairs. Additional analyses show that HarnessLLM exhibits better generalizability and benefits the code generation performance with test-time scaling.

---

[4]An output verifier can use a combination of strategies, so the numbers do not add up to 100.

## Limitations

While our work proposes a new test harness generation framework that enables more diverse test case synthesis beyond input-output checks, it has two main limitations: ❶ Dependence on ground-truth programs. Our method assumes access to a correct reference implementation, which is used to compute reward signals during training. However, this assumption may not hold in real-world settings where the ground-truth code solution is unavailable or difficult to collect. Future work could explore directions such as using weaker oracles or generalizing models trained on simpler tasks with ground truths to more difficult tasks. ❷ Limited scenarios. Our work focuses on improving LLMs' core ability to generate test cases. To study this, we evaluate on function-level test case generation as an abstraction of the problem. However, although we evaluate on broad benchmarks, this task only represents a subset of scenarios for LLM-based automatic debugging. Future work could extend this to real-world repository-level debugging. For example, recent works on repository-level debugging often adopt an agentic, multi-stage framework, where specialized agents are created for particular sub-tasks in the whole pipeline Tang et al. (2025); Chen et al. (2024b); Lops et al. (2025). *E.g.,* some agents first locate candidate code units responsible for the bugs, and other agents then generate test cases targeting each localized unit. Within the context of each individual code unit, the test generation reduces to a similar setting studied in this paper.

Despite these limitations, our results demonstrate the effectiveness of the proposed harness generation framework. We believe this framework can serve as a foundational component for more complex test generation and debugging systems, such as repository-level codebase testing.

## Broader Impact

This work aims to enhance the reliability and robustness of AI-generated programs by developing improved methods for testing and debugging. However, while our method shows clear improvements over the baseline, it does not capture all bugs or provide any guarantees on the program's correctness. Our experiments show that some bugs remain hidden and some correct programs may be mistakenly flagged. Therefore, users should remain cautious when interpreting the execution results of our generated test cases. We advise that any use of this system in high-stakes environments be accompanied by additional verification and human oversight.

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

# A  Dataset Construction

## A.1  Training Data

To train LLMs for test case generation, we collect data in the triplets of problem description $q$, buggy program $f$, and ground-truth program $g$. We consider Python programs in this paper. We source such triplets from existing coding datasets, including TACO Li et al. (2023), SYNTHETIC-1 Intellect (2025), LeetCode Xia et al. (2025), and Codeforces MatrixStudio (2025). These datasets come with the problem description, a ground-truth program, and a list of ground-truth test cases. We use the following three steps to collect data:

❶ Filter ground-truth programs: We run the given ground-truth program $g$ on all test cases and only keep problems where $g$ passes all test cases.

❷ Generate buggy programs: We sample candidate programs from `Qwen2.5-Coder` 1.5B, 3B, 7B Hui et al. (2024), and `DeepSeek-R1-Distill-Qwen-1.5B` DeepSeek-AI (2025). We sample 8 programs from each model and run the programs on all ground-truth test cases. We only keep programs that pass at least one test case but not all test cases, resulting in partially correct programs. If there are multiple candidates that satisfy the requirement, we use the two that pass the most test cases, which makes it harder to find bugs.

❸ Decontamination: We decontaminate training data against all evaluation benchmarks based on the problem description.

We use all collected data for RL training and a subset of data for SFT, ensuring that models see new data during RL training. Table 6 shows the statistics of our training set. Specifically, the dataset contains two types of problems: standard input/output problems that read from `stdin` and return to `stdout`, as well as functional problems that implement a function in Python. Since the number of functional problems is small, we create two versions for each functional problem, where one contains a few example input-output pairs in the description, and the other does not.

Table 6: Statistics of our training data.

|  | Statistic |
| --- | --- |
| # triplets for RL | 12,043 |
| # unique problems for RL | 7,748 |
| # triplets for SFT | 6,805 |
| # unique problems for SFT | 4,383 |
| # responses for SFT | 15,619 |

**SFT Data.**  To collect SFT data, we use the rejection sampling technique Touvron et al. (2023). Specifically, we prompt `Qwen3-32B` to generate 6 responses for each pair of description and buggy program. Figures 8 and 9 show the prompt we use for input-output testing and test harnesses, respectively. Particularly, for harness generation, we encourage the model to use diverse strategies to validate outputs, such as checking specific invariants and comparing with a brute-force solution, which is similar to the strategy used in prior works Zhang et al. (2023b). We run generated test cases on both ground-truth program $g$ and buggy program $f$ and only keep responses where $g$ passes the test but $f$ does not. We keep the amount of SFT data the same for input-output testing and harness testing.

## A.2  Evaluation Data

We evaluate on three popular code generation datasets: MBPP+ Austin et al. (2021); Liu et al. (2023), LIVECODEBENCH Jain et al. (2025), and CODEFORCES Penedo et al. (2025). Although these datasets are designed for code generation tasks, we convert them into bug-finding tasks following the procedure in Section A.1.

Specifically, for LIVECODEBENCH, we use problems from 2024/10 to 2025/4. For CODEFORCES, we use samples in the test split. For both datasets, we use correct public submissions as the ground-truth program, after rerunning and filtering the submissions on all test cases.

Table 8: True bug rate (higher is better) of input-output-based testing with `Qwen3-32B` when only evaluating the first $k$ generated test cases. We use the SEEN version of LIVECODEBENCH and CODEFORCES.

|       | MBPP+ | LIVECODEBENCH | CODEFORCES |
|-------|-------|---------------|------------|
| $k = 1$  | 49.3 | 54.8 | 67.1 |
| $k = 3$  | 59.0 | 54.6 | 53.2 |
| $k = 5$  | 57.4 | 53.6 | 42.5 |
| $k = 10$ | 54.6 | 51.3 | 39.5 |

Table 9: True bug rate (higher is better) of test harnesses with `Qwen3-32B` when only evaluating the first $k$ generated test cases. We use the SEEN version of LIVECODEBENCH and CODEFORCES.

|       | MBPP+ | LIVECODEBENCH | CODEFORCES |
|-------|-------|---------------|------------|
| $k = 3$  | 66.6 | 48.5 | 57.9 |
| $k = 5$  | 68.6 | 55.1 | 54.8 |
| $k = 10$ | 67.7 | 53.8 | 48.6 |
| $k = 20$ | 67.3 | 53.3 | 44.2 |

For MBPP+, we directly use the split MBPP+FIX (HARD) in `UTGen-32B` Prasad et al. (2025), which is collected similarly to the above procedure. Particularly, we notice the problem descriptions in MBPP+ are overly simplified and without clear input specifications (*e.g., 'Write a function to find the length of the longest palindromic subsequence in the given string'*, without specifying that the input string should be non-empty). We thus use `Qwen3-32B` to add an input specification to the problem (detailed prompt in Figure 10). To make sure the ground-truth program $g$ matches the description after modification, we further prompt `Qwen3-32B` to adapt the original $g$ to the new description (detailed prompt in Figure 11). Finally, we filter the modified ground-truth programs and only keep those that pass the original ground-truth test cases.

Table 7: Statistics of evaluation datasets.

|                          | # data |
|--------------------------|--------|
| MBPP+FIX (HARD)          | 141    |
| LIVECODEBENCH SEEN       | 76     |
| LIVECODEBENCH UNSEEN     | 93     |
| CODEFORCES SEEN          | 100    |
| CODEFORCES UNSEEN        | 84     |

Table 7 lists the statistics of all evaluation benchmarks.

## B   Implementation Details

### B.1   Number of Test Cases

For the teacher model and SFT models, we observe that the number of test cases in a response significantly affects the final performance. For example, although we allow models to generate multiple test cases in each response, Tables 8 and 9 show that the performance of `Qwen3-32B` can vary significantly if we only evaluate the first $k$ test cases. Both methods' performance improves as we evaluate on fewer test cases, especially for input-output-based testing. This confirms the observations in Figure 3, where the performance of input-output testing quickly drops when generating more test cases. Based on these results, for the teacher model and SFT models of input-output testing, we report the performance when $k = 1$. For test harnesses, we report the performance when $k = 5$.

For the RL models, we observe that the models automatically find a good number of test cases to generate. For instance, the RL trained `Qwen3-4B` model for input-output testing generates 1.96 test cases in each response on average. Thus, we allow the model itself to determine the number of test cases, and we only restrict the maximum test cases at 20.

### B.2   Training Hyperparameters

We run all experiments on 16 NVIDIA H100 GPUs. The RL training for our model takes around 1,500 GPU hours. The RL training for the input-output baseline takes around 1,150 GPU hours. Table 10 lists the hyperparameters for SFT and RL training. Note that we use the same hyperparameters for all models.

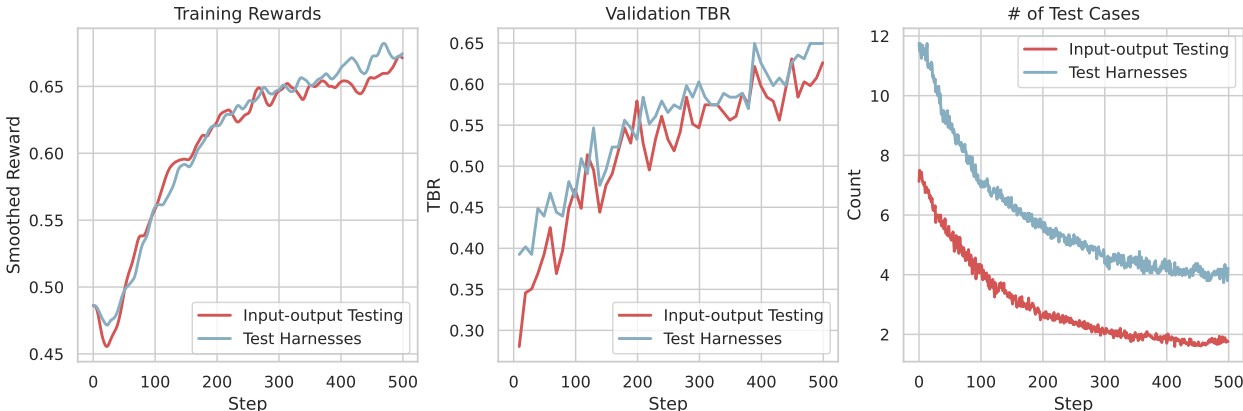

Figure 6: Dynamics of smoothed training rewards, true bug rage (TBR ↑) on validation set, and the number of generated test cases throughout RL training.

Table 11: Performance of `Llama3.2-3B` on finding bugs (average of 8 runs). `I/O`: input-output testing. `Har`: test harnesses.

| | MBPP+Fix (Hard) | | | LCB Seen | | | CF Seen | | | LCB Unseen | | | CF Unseen | | |
|---|---|---|---|---|---|---|---|---|---|---|---|---|---|---|---|
| | GI ↑ | ITR ↓ | TBR ↑ | GI ↑ | ITR ↓ | TBR ↑ | GI ↑ | ITR ↓ | TBR ↑ | GI ↑ | ITR ↓ | TBR ↑ | GI ↑ | ITR ↓ | TBR ↑ |
| RL (I/O) | 71.3 | 37.3 | **45.3** | 47.0 | 42.3 | 30.3 | 67.6 | 53.9 | **32.8** | 21.0 | 61.8 | 8.3 | 37.4 | 66.2 | 10.1 |
| RL (Har) | **77.9** | 37.3 | 42.6 | **76.5** | **33.9** | **31.4** | **81.4** | **43.8** | 29.9 | **59.9** | **39.9** | **17.7** | **73.4** | **43.3** | **21.6** |

### B.3 Classifying Testing Strategies

We prompt `Qwen3-32B` to identify specific testing strategies used by our model. Specifically, given the generated harness code, we ask the model to identify strategies used in each input generator and output verifier. The detailed prompts are listed in Figures 12 and 13.

## C Additional Results

### C.1 Training Dynamics

Figure 6 shows the dynamics of RL training for both methods. As can be observed, our method consistently achieves a higher TBR on the validation set than the baseline. Moreover, both methods generate fewer test cases as the training progresses, approaching the optimal number in Section B.1. This indicates that the models are learning the best number of test cases for generation.

### C.2 Results on Llama

Table 10: Training hyperparameters. The same hyperparameters are used for all models.

| SFT Training | |
|---|---|
| # Epochs | 15 |
| Batch size | 96 |
| Learning rate | $1e{-}5$ |
| LR scheduler | cosine |

| RL Training | |
|---|---|
| # Steps | 500 |
| Batch size | 128 |
| # Rollouts per question | 8 |
| Learning rate | $1e{-}6$ |
| LR scheduler | None |
| Max response length | 16,384 |

Table 11 shows the performance when training on `Llama3.2-3B` model. As can be observed, our model for test harnesses achieves comparable performance with input-output testing on the SEEN version of the datasets. However, it significantly outperforms the input-output testing when evaluated on the UNSEEN version, *e.g.,* a relative improvement over 110% in TBR on LIVECODEBENCH. The results indicate that input-output testing has the risk of overfitting to a particular distribution of bugs, whereas test harnesses have better generalizability.

Table 12: Line and branch coverage on ULT.

| | LCov@4 ↑ | BCov@4 ↑ | LCov@16 ↑ | BCov@16 ↑ |
|---|---|---|---|---|
| RL (I/O) | 66.32 | 80.67 | 73.83 | 84.56 |
| RL (Har) | **70.38** | **82.20** | **78.41** | **87.14** |

Table 13: Comparison with `TrickCatcher` on the unseen version of test data.

| | LiveCodeBench | | | Codeforces | | |
|---|---|---|---|---|---|---|
| | GI ↑ | ITR ↓ | TBR ↑ | GI ↑ | ITR ↓ | TBR ↑ |
| TrickCatcher (frozen) | 25.9 | 42.9 | 8.1 | 31.1 | 46.3 | 5.2 |
| Harness (frozen) | 28.9 | 44.6 | **8.3** | 29.2 | 50.1 | **6.1** |
| Harness (RL-trained) | 51.1 | 17.7 | **37.2** | 67.3 | 26.8 | **37.9** |

## C.3   Results on ULT

We evaluate on the officially released lite subset of ULT Huang et al. (2025), which covers challenging real-world coding problems. Since the evaluation code is not available, we only report the LCov@k and BCov@k, which measure the line coverage and branch coverage of the first $k$ generated test cases. For fair comparison, we measure the coverage under the same number of unique test cases for both methods. Table 12 shows the results for the two RL-trained `Qwen3-4B` models. As can be observed, our method consistently achieves higher line and branch coverage than the baseline. The results suggest that the improvements of our method can be generalized beyond the competitive programming tasks.

## C.4   Comparison with `TrickCatcher`

We additionally compare with `TrickCatcher` Liu et al. (2025), which uses the LLM to first generate multiple repaired programs of the program under test (PUT). It then generates an input generator, which is executed to produce test inputs. Finally, it collects test cases where the PUT produces different outputs from the repaired programs and uses the majority output of the repaired program as the expected output.

Table 13 shows the performance on the unseen version of test data when using `Qwen3-4B` as the base model. We observe that `TrickCatcher` achieves performance comparable to our harness generation when both rely on a frozen base model. However, `TrickCatcher` incurs substantially higher computational cost: for each sample, it generates multiple repaired program variants (four in our implementation) and an input generator, resulting in four additional LLM queries per task than our method. In contrast, our method generates a complete harness in a single model call. The results show that our method is more efficient and achieves better performance.

## C.5   Generalization to Buggy Programs from `gpt-oss`

In Section 4.2, to evaluate our models' generalizability, we evaluate on buggy programs generated by `Qwen3-14B`, which is never seen during training. In addition to that, we also repeat the same experiment with buggy programs generated by `gpt-oss-20b`. The results in Table 14 demonstrate that our method still consistently outperforms the baseline method, indicating that the observed improvements can generalize beyond the models seen during training.

## C.6   Diversity of Generated Test Cases

To measure the diversity of generated test cases, we calculate three metrics for the same number of test inputs generated by our method and the input-output testing baseline. ① Unique ratio: We calculate $\frac{\text{\# of unique inputs}}{\text{\# of total inputs}}$, where equality is defined by string matching. ② Length range: We calculate

Table 14: Performance on buggy programs generated by `gpt-oss-20b`.

|  | LiveCodeBench | | | Codeforces | | |
|---|---|---|---|---|---|---|
|  | GI ↑ | ITR ↓ | TBR ↑ | GI ↑ | ITR ↓ | TBR ↑ |
| I/O (RL) | 24.1 | **23.1** | 19.6 | 43.5 | 29.3 | 28.9 |
| Harness (RL) | **36.9** | 23.9 | **22.4** | **63.7** | **26.6** | **30.5** |

Table 15: Bug-finding performance of `Qwen3-4B` when controlling the number of test cases. `I/O`: input-output testing. `Har`: test harnesses.

|  | MBPP+Fix (Hard) | | | LCB Seen | | | CF Seen | | | LCB Unseen | | | CF Unseen | | |
|---|---|---|---|---|---|---|---|---|---|---|---|---|---|---|---|
|  | GI ↑ | ITR ↓ | TBR ↑ | GI ↑ | ITR ↓ | TBR ↑ | GI ↑ | ITR ↓ | TBR ↑ | GI ↑ | ITR ↓ | TBR ↑ | GI ↑ | ITR ↓ | TBR ↑ |
| | | | | | | | Repeat Once | | | | | | | | |
| RL (I/O) | 80.5 | 11.6 | 73.8 | 67.4 | 9.7 | 65.1 | 87.8 | 16.2 | 75.6 | 36.0 | 18.7 | 33.9 | 49.1 | 30.7 | 34.2 |
| RL (Har) | 83.6 | 12.1 | **74.9** | 75.0 | 7.9 | **68.4** | 90.2 | 16.1 | **76.9** | 43.3 | 13.6 | **37.1** | 58.3 | 21.4 | **34.5** |
| | | | | | | | Repeat 5 Times | | | | | | | | |
| RL (I/O) | 86.7 | 22.3 | 69.5 | 79.9 | 23.7 | 63.2 | 97.2 | 34.8 | 63.8 | 54.8 | 46.5 | 31.7 | 75.0 | 56.2 | 31.8 |
| RL (Har) | 83.8 | 12.3 | **75.0** | 75.0 | 7.9 | **68.4** | 90.8 | 16.1 | **77.4** | 49.5 | 14.5 | **40.9** | 61.9 | 22.6 | **34.5** |
| | | | | | | | Repeat 20 Times | | | | | | | | |
| RL (I/O) | 88.7 | 34.0 | 60.3 | 86.8 | 42.1 | 51.3 | 100.0 | 56.0 | 44.0 | 64.5 | 72.0 | 18.3 | 86.9 | 79.8 | 16.7 |
| RL (Har) | 83.5 | 12.5 | **74.6** | 77.6 | 7.9 | **71.1** | 90.8 | 16.4 | **77.1** | 52.2 | 14.9 | **42.5** | 63.1 | 23.8 | **35.7** |

$\log(\text{max\_length} + 1) - \log(\text{min\_length} + 1)$, where max_length and min_length are the maximum and minimum lengths of the inputs. ③ Length std: We calculate the standard deviation of the log of each input length. For all metrics, we compute the value for each individual problem and take the average over all problems. Results in Table 5 show that our method generates more diverse inputs and inputs with various lengths than the baseline.

## C.7 Performance under Controlled Number of Test Cases

Our experiments in Section 4.2 demonstrate that different methods should generate different numbers of test cases for the best performance. Particularly, for input-output testing, models usually have better performance when generating fewer test cases, since more test cases lead to a higher probability that one of the test cases is wrong. By contrast, for test harnesses, performance can be further improved by scaling up the number of test cases, which increases the test coverage without sacrificing accuracy. Nevertheless, in the following section, we also report the performance when controlling the number of test cases.

Specifically, we repeat the experiments in Tables 1 and 2 but instruct the input-output testing model to only generate a single test case in each response. For test harnesses, we also prompt the model to generate a single input generator. Then, to get more test cases, we sample multiple responses for input-output testing and run the same input generator multiple times with different random seeds for test harnesses. Table 15 presents the performance under this controlled setting. As can be observed, our method consistently outperforms input-output testing, and the gap becomes larger when increasing the number of test cases. These results demonstrate the consistent improvements of the proposed test harness.

Moreover, we also rerun the test-time scaling experiment in Table 3. Here, we further restrict each input generator to have a single test input, thus ensuring the two methods have the same number of test cases for each candidate program. For both methods, we sample 5 responses for each candidate program to obtain more test cases. The results in Table 16 show that our method surpasses the baseline in most settings. Particularly, it significantly outperforms the baseline when the strongest `Qwen3-32B` is used as the code generator, demonstrating the superior generalizability and potential weak-to-strong generalization of our method.

---

***q***: You are given an integer array nums. Select a subarray of nums such that: (1) All elements in the subarray are unique. (2) The sum of the elements in the subarray is maximized. Return the maximum sum of such a subarray.

```python
def generate_input_1():
    """Hardcoded test case"""
    return [{"nums": [-5,-3]}]

def generate_input_2():
    """Dynamically generates test cases"""
    input_list = []
    for _ in range(3):
        length = random.randint(1, 10)
        nums = [random.randint(-100, -1)
                for _ in range(length)]
        input_list.append({"nums": nums})
    return input_list
```

```python
def check_output(input: Dict[str, Any], output: Any):
    """Validates output for all cases"""
    nums = input["nums"]
    if nums == [-5,-3]:
        assert output == -3
    else:
        all_negative = all(n < 0 for n in nums)
        if all_negative:
            expected = max(nums)
            assert output == expected
```

Figure 7: A sample output using a combination of strategies for input generators and output verifier.

## C.8 Sample Outputs

By programmatically generating inputs and validating outputs, our method enables more diverse testing strategies. Figure 7 illustrates an example where our model uses a combination of testing strategies.

Table 16: Best-of-8 performance on LIVECODEBENCH where the code is selected based on the execution results of the generated test cases.

|  | Code Generator | | |
|---|---|---|---|
|  | Qwen3-4B | Qwen3-14B | Qwen3-32B |
| Original pass@1 | 52.60 | 60.23 | 63.53 |
|  | **1 test case per program** | | |
| RL (I/O) | 60.41 | **65.10** | 65.98 |
| RL (Harness) | **60.70** | 64.81 | **68.33** |
|  | **5 test cases per program** | | |
| RL (I/O) | **61.88** | 67.16 | 68.04 |
| RL (Harness) | 61.00 | **67.74** | **72.14** |

```
Given a problem statement and a Python program that aims to solve it, your task is to **
write test cases** that uncover any potential bugs.

### **Task Overview**

You should output a JSON object that contains a list of test cases for the provided program
. Each test case should include:
1. **input_str**: The exact text to feed into stdin.
2. **expected_output**: The exact text the program should print.

We will run each test by feeding `input_str` into the program and comparing its stdout
against `expected_output`.

### **Required Format**

```json
[
  {
    "input_str": "input 1",
    "expected_output": "output 1"
  },
  {
    "input_str": "input 2",
    "expected_output": "output 2"
  }
  // ... up to 20 test cases total
]
```

### **Constraints**

* Generate **1-20** test cases.
* Don't include comments or extra fields in the JSON.
* Each input_str and expected_output must be a valid JSON string.

The problem is as follows:
{description}

And the program is as follows:
```python
{target_code}
```
```

Figure 8: Prompt used for input-output-based testing. Note that this prompt assumes the program reads input from stdin.

Given a problem statement and a Python program that aims to solve it, your task is to **
write a test harness** that uncovers any potential bugs.

### **Task Overview**

You will deliver **a single** code block to define functions that can be run by our
framework to generate inputs, run the program, and validate its outputs.
Consider two categories of test cases:
- **Hardcoded cases**: Manually crafted input-output pairs that expose known or likely bugs
.
- **Dynamic cases**: Programmatically generated inputs that stress-test the implementation
(e.g., randomized, combinatorial, large or edge-case inputs).

### **Required Functions**

```python
from typing import List

def generate_input_1() -> List[str]:
    """
    Return between 1 and 4 valid input strings, each a complete stdin payload for the
    target program.
    Consider the following strategies:
      - Manually craft inputs that expose bugs.
      - Dynamically generate randomized, combinatorial, large, or edge-case inputs for
      stress testing.
    """
    # Your code here
    return input_list

def generate_input_2() -> List[str]:
    """
    Another function to return between 1 and 4 valid input strings.
    Employ a different strategy than previous input generation functions.
    """
    # Your code here
    return input_list

# You may add up to 3 more functions named generate_input_3(), generate_input_4(), etc.

def check_output(generated_input: str, captured_output: str) -> None:
    """
    Validate the output for a single generated input.
    Inputs:
        - generated_input: The input string passed to the target program.
        - captured_output: The exact stdout produced by the target program.

    Hints: When exact outputs are hard to predict, avoid asserting them. Instead, consider:
      - Check key properties or invariants, e.g., output is sorted, has correct length,
      matches a pattern, has correct value ranges, etc.
      - Compare against a simple brute-force implementation
    """
    # Your code here
```

### **Execution Flow**

1. The framework calls generate input functions to obtain a list of test strings.
2. For each string:
    * It runs the target program with that string on stdin.
    * Captures stdout into `captured_output`.
    * Calls `check_output(generated_input, captured_output)`.
3. If any assertion fails, the test suite reports an error.

### **Constraints**

* Provide one contiguous block of Python code that defines all required/optional functions.
 Do not invoke the functions yourself-only define them.
* Define up to 5 input generation functions, each returning between 1 and 4 inputs.

```
* The dynamic input functions must employ diverse strategies to generate inputs. Avoid
generating inputs with the same logic or from the same distribution.
* Runtime limit per check_output call: 5 seconds.

The problem is as follows:
{description}

And the program is as follows:
```python
{target_code}
```
```

Figure 9: Prompt used for test harnesses generation. Note that this prompt assumes the program reads input from stdin.

```
Given the following coding problem and a corresponding solution, improve the problem
description by adding input specifications. Include details such as:
- Valid input types (e.g. "integer", "string", "list of floats").
- Reasonable value ranges (e.g. "0 <= n <= 1000").
- Format constraints (e.g. "no empty strings", "no null/None values").

Do not change the original requirements or add example cases, just append the
specifications.

Problem:
{problem}

Code:
```python
{code}
```
```

Figure 10: Prompt used for adding input specifications on MBPP+.

```
Given the following coding problem and a corresponding solution, decide whether the
solution contains a bug or not. If yes, rewrite the code to fix the bug. Remember to look
for edge cases where the code fails to handle.

Problem:
{problem}

Code:
```python
{code}
```

Output your answer in the following format:
```python
fixed_code
```
where fixed_code is the rewritten code that fixes the bug. If the code is correct, just
return the original code without any changes.
```

Figure 11: Prompt used for adapting the ground-truth programs to the new descriptions on MBPP+.

```
Given the following code snippet for a test harness, determine the strategy used in each `
generate_input` function.

Code:
```python
{code}
```

Select from the following options:
- hardcoded: the function returns hardcoded inputs.
- dynamic: the function generates inputs dynamically, e.g., random sampling, or
combinatorial generation.

Think about the code step by step and then output your final answer in the following format
:
```json
<used strategies>
```
where <used strategies> is a list of the strategies used in each function.

Notes:
- The list should have the same length as the number of `generate_input` functions in the
code.
- If a function uses a combination of the above strategies, select the dominant strategy.
```

Figure 12: Prompt used for identifying strategies in input generators.

```
Given the following code snippet for a test harness, determine the strategies used in the `
check_output` function.

Code:
```python
{code}
```

Select from the following options:
- reference implementation: the function compares the output with a reference
implementation, e.g., a brute-force solution, or a correct implementation.
- invariant checking: the function checks whether the output satisfies certain invariants
or properties, e.g., whether the output is sorted, or whether the output has valid types
and lengths.
- hardcoded: the function compares the output with hardcoded expected outputs.

Think about the code step by step and then output your final answer in the following format
:
```json
<used strategies>
```
where <used strategies> is a list of the strategies used in the function.

Notes:
- If the function uses a combination of the above strategies, return a list containing all
the strategies used, e.g., ["reference implementation", "invariant checking"].
- If the function does not contain any of the above strategies, return an empty list [].
```

Figure 13: Prompt used for identifying strategies in the output verifier.

