# OpenReview forum: "HarnessLLM: Automatic Testing Harness Generation via Reinforcement Learning"
_TMLR — Under review for TMLR_

### Review · Reviewer_Uoei · 2026-05-19

**Summary Of Contributions:**

The paper proposes HarnessLLM, a two-stage training pipeline that trains LLMs to generate executable test harnesses rather than (input, expected_output) pairs for the purpose of debugging buggy code. A harness in this framework consists of one or more input-generator functions plus a check_output validator that can encode invariants, comparisons against a reference implementation, or hardcoded expected outputs. The motivation is the observation that input-output test generation is bottlenecked by the model's ability to predict correct outputs, when the program under test is harder than the tester can independently solve, the paradigm collapses.

The harness reformulation is well-motivated, and the scaling-with-test-cases result (Figure 3) provides a qualitative and quantitative justification.The test-time-scaling application (Table 3), in which a small harness model is used to select the best-of-8 outputs from a much larger code generator. The strategy-distribution analysis (Figure 5) showing that the trained model adapts its testing strategy to problem difficulty and to the correctness profile of the buggy program is a nice qualitative finding.

**Audience:**

Yes

**Audience Explanation:**

The reformulation of LLM-based test generation from input-output pairs to executable harnesses is a real conceptual contribution that opens up validator-design choices (invariant checking, brute-force reference comparison) that the I/O paradigm structurally cannot express. The test-time-scaling result, where a 4B harness generator improves a 32B code generator's pass@1 on LiveCodeBench by ~6 points, is the kind of weak-to-strong-style result that is currently of broad interest in the LLM-for-code subcommunity. Even allowing for the concerns above, the underlying ideas are worth the TMLR audience's attention if the paper is revised to address them.

**Claims And Evidence:**

No

**Claims Explanation:**

1. **The "strong coding LLM" framing rests on a single model from a single family at one size**. The introduction states that "even the strong coding LLMs are not inherently capable of test code generation" and uses this finding to motivate the entire training pipeline. The empirical support in Table 1 consists exclusively of Qwen3-32B in two prompting modes. Llama3.2-3B (the only non-Qwen base model in the paper) appears only in Appendix C.2 and only at 3B scale. gpt-oss-20b appears only as a source of buggy programs in Appendix C.5, not as a tester. There is no frontier closed-source model (e.g., Claude Opus, GPT-4-class, Gemini 2.x) tested even at the prompting level. Without at least one non-Qwen large model evaluated under the same conditions as the Qwen3-32B row, the broad framing "off-the-shelf strong LLMs cannot do this" is not adequately substantiated; the evidence supports a narrower claim about Qwen3-32B specifically.

2. **The MBPP+Fix Hard comparison to UTGen-32B is not apples-to-apples.** Appendix A.2 reports that for MBPP+Fix Hard, the authors used Qwen3-32B to rewrite each problem description with explicit input specifications and then re-adapted the ground-truth solution to match the new descriptions. This is a meaningful methodological difference that should be flagged in the main text and the Table 1 caption, not buried in the appendix. Without that disclosure, the apparent UTGen-32B vs Qwen3-32B comparison reads as a head-to-head when it is in fact a comparison across slightly different benchmark variants.

3. **No frozen Qwen3-4B baseline is reported in Table 1**. Every Qwen3-4B row in every table in the paper is after at least one stage of training (SFT, RL, or SFT+RL). The relevant ablation cell, frozen Qwen3-4B prompted in both I/O and harness modes on the same problems, is missing. This makes it impossible to read off how much of the trained model's performance comes from the harness format versus the training procedure. The numbers are inexpensive to obtain (inference only, no training) and would directly address the attribution question.

4. **Error-trace-based iterative refinement is never evaluated.** The paper moves directly from "off-the-shelf LLMs struggle to write effective harnesses" (Section 3.3) to "therefore we train them with SFT+RLVR." It does not test intermediate alternatives. A substantial body of prior work {UTFix (Rahman et al., 2025), ChatUniTest (Chen et al., 2024), Self-Edit (Zhang et al., 2023a)} has shown that iterative refinement using execution feedback (failure messages, compilation errors, assertion violations on a reference implementation) can substantially improve test generation from frozen LLMs. Before concluding that fine-tuning is necessary, the paper should evaluate a baseline in which a frozen Qwen3-32B (and ideally Qwen3-4B) generates an initial harness, runs it against the ground-truth program, observes compilation and assertion failures, and iteratively refines its harness over k turns. If that closes the gap, the case for RL training weakens substantially. If it does not, the case for training is much stronger and the paper benefits.

5. **The C++ generalization claim on TrickyBugs is overstated.** Table 4 reports GI 33.0 vs 28.4 (+4.6), ITR 16.1 vs 14.6 (worse by 1.5), and TBR 27.1 vs 26.6 (+0.5). The paper frames this as evidence that "our training improves the general capability to generate test harnesses for diverse programming languages." But TBR, the metric that measures actual bug detection, improves by 0.5 absolute points on 250 problems averaged across 8 runs with no variance reported. ITR (the rate of invalid tests) actually gets worse. The honest reading is that input-generation capability (GI) transfers to C++ but bug-detection performance (TBR) does not improve meaningfully. The framing should be qualified accordingly, or additional C++ benchmarks should be added.

6. **The Llama3.2-3B MBPP+Fix Hard anomaly is unexplained.** Table 11 shows Llama3.2-3B RL-Harness achieving 42.6 TBR on MBPP+Fix Hard versus 45.3 TBR for Llama3.2-3B RL-I/O. This is the only cell in any main-experiment trained-model row where the harness paradigm loses to the I/O paradigm.

7. **Table 2 omits MBPP+Fix Hard**. The unseen-bug-generator generalization study includes LiveCodeBench and Codeforces but not MBPP+Fix Hard. Even if MBPP+Fix's buggy corpus is a fixed external set rather than newly generated, including it (or explaining its omission) would round out the cross-benchmark story.

**Requested Changes:**

1. **Substantiate or soften the "strong coding LLMs" framing**. Either (a) add at least one frontier closed-source baseline (Claude Opus, GPT-4-class, or Gemini 2.x) to Table 1, evaluated zero-shot in both I/O and harness prompting modes; or (b) add a 32B-scale non-Qwen open-source baseline (e.g., Llama 3.x at a comparable scale, or DeepSeek-Coder-33B) under the same conditions. If neither is feasible, rephrase the abstract and introduction to scope the framing to Qwen-family models and explicitly acknowledge that cross-family generalization is unverified.

2. **Disclose the MBPP+Fix Hard modification in the main text and Table 1 caption.** Section 4.1 or the Table 1 caption should explicitly state that the MBPP+Fix Hard split used is a Qwen3-32B-rewritten variant of UTGen's release, not the original

3. **Add the frozen Qwen3-4B baseline.** Report Qwen3-4B (no training) numbers in both I/O and harness prompting modes on MBPP+Fix Hard, LiveCodeBench Seen, and Codeforces Seen.

4. **Add an error-trace / iterative-refinement baseline before claiming fine-tuning is necessary.**

5. **Qualify or strengthen the C++ generalization claim**

6. **Include MBPP+Fix Hard in Table 2 or justify its omission**

7. Analyze the Llama3.2-3B MBPP+Fix Hard anomaly. Discuss why the trained harness model loses to the trained I/O model in this single cell (42.6 vs 45.3 TBR).

---

### Review · Reviewer_RWu6 · 2026-06-12

**Summary Of Contributions:**

In this paper, the authors propose a model (trained via RLVR) to generate harness that can create test cases at scale to identify bugs in a code. The main contribution of the work is to propose the reward model and the idea of an harness for testing framework instead of generating pairs of (input, expected output)

**Additional Comments:**

None

**Audience:**

Yes

**Audience Explanation:**

Yes, the problem is important

**Claims And Evidence:**

Yes

**Claims Explanation:**

Yes, experiments are extensive

**Requested Changes:**

I have several doubts/questions regarding the solution outline

1. "even the strong coding LLMs are not inherently capable of test code generation" - Why should test code generation be such a hard task? There are no experiments to prove this statement clearly or citations to existing work. We need qualitative examples for this particular claim as well

2. "Rather than letting the language model directly generate input-output pairs, we prompt it to write executable code, a matching rival to the testee, to generate test inputs and validate target program outputs." - Why is this an easier task? Even here, for this paper, the authors needed to train via RLVR. I am wondering why an RLVR based set up cannot be done for (input, expected output) pairs?

It is a very tall claim that both drawbacks are addressed simultaneously through this approach

3. I am still unsure of the intuition that it is easier to generate verifier code - "If the target program cannot generate the correct output, why can the executor generate the correct verified code?" Even in the experiments, it seems that the same model is not being used for both. Target programs are generated by one model and verifier code seems to be generated by another model. Is this correct or not?

4.  In the introduction, the authors need to highlight that the target program has subtle bugs as compared to the ground truth. It is not clear if the target program used for the SFT data are hard negatives or soft negatives.

5. Very importantly, how does RL training help? Which part does it improve - is it the input generation part of the code for validating the output?

6.Why the smaller model is finetuned? Why not the bigger model? Is it because of lack of resources? Why generate the target harnesses from a larger model then?

---

### Review · Reviewer_21Dq · 2026-07-10

**Summary Of Contributions:**

The paper proposes HarnessLLM, a two-stage method for training LLMs to generate executable test harnesses rather than plain input-output test pairs. The generated harnesses contain input generators and a `check_output` function that can validate outputs through exact checks, reference implementations, or invariants. The training pipeline first uses rejection-sampled SFT data from Qwen3-32B, then applies RLVR with a reward based on whether a generated harness is passed by a ground-truth implementation and failed by a buggy implementation.

## Strengths:

- The core idea is useful and well motivated: programmatic test generation can express richer input distributions and weaker-but-useful oracles than fixed input-output pairs.
- The RL reward is simple, executable, and well aligned with the bug-finding objective.
- The paper evaluates on several bug-finding settings, including MBPP+Fix, LiveCodeBench, Codeforces, ULT, TrickyBugs, unseen bug generators, and test-time code selection.
- The ablations on number of test cases are helpful and support the claim that harness-based testing scales better than hardcoded input-output tests.

## Weaknesses:

- The novelty claim is overstated. The paper is related to property-based testing, parameterized unit tests, oracle/verifier generation, fuzzing, symbolic/concolic test generation, and LLM-assisted test generation. The paper should position its contribution as RL training for LLM-generated harnesses, not as the first LLM-based harness/test-oracle generation paradigm.
- The main empirical evidence is promising but not yet fully convincing because the evaluation sets are small, mostly derived from synthetic LLM-generated bugs, and lack confidence intervals or significance tests.
- The comparison to baselines needs stronger cost and budget controls. The harness method can reuse randomized generators, while the input-output baseline is prompted to emit more fixed tests; this difference is part of the method, but the paper should still report matched execution counts, token budgets, samples, and wall-clock cost.
- The “debugging information” claim is not empirically evaluated. The paper shows better bug-finding rates, but not whether generated assertion failures are understandable or useful to developers.
- The strategy-diversity analysis relies on Qwen3-32B classification of generated harnesses, but no validation of this classifier is reported.

**Audience:**

Yes

**Audience Explanation:**

The paper targets an important problem at the intersection of LLMs, code generation, automated testing, and RL with verifiable rewards. TMLR readers interested in code LLMs, program synthesis, test-time scaling, and executable feedback would likely find the main result interesting: training a small model with SFT plus RLVR to write harnesses can outperform a larger prompted model and an input-output-trained counterpart.

The work is also timely because many code-generation systems already use generated tests for selection or debugging, and this paper studies a richer test representation. The finding that harnesses can scale the number of generated tests more safely than fixed input-output pairs is valuable.

**Claims And Evidence:**

No

**Claims Explanation:**

The paper provides a substantial empirical case that RL-trained harness generation can outperform RL-trained input-output generation on the authors’ constructed bug-finding benchmarks. The gains in TBR on MBPP+Fix, LiveCodeBench, and Codeforces are meaningful, and the test-time scaling results are useful.

However, several claims are not yet supported strongly enough:

- The claim that HarnessLLM is the first LLM-based testing harness generation method is too broad. The approach overlaps with LLM-generated property-based tests [3], oracle/verifier-guided code synthesis [4], generated tests for code selection [1], LLM-powered fuzzing [9], and traditional property-based or parameterized test harnesses [5,6].
- The claim of broad generalization is only partially supported. The “unseen model” experiments still use synthetic LLM-generated bugs, and the C++ TrickyBugs result shows only a small TBR improvement over the input-output baseline.
- The claim that harnesses provide better debugging information is plausible but not measured. The paper should evaluate whether the generated checkers localize failures, produce interpretable assertions, or help a developer fix bugs.
- The evidence is weakened by small evaluation set sizes, such as 76 LiveCodeBench Seen examples and 100 Codeforces Seen examples, without confidence intervals or statistical tests.

**Requested Changes:**

Critical to acceptance:

- Narrow the novelty claim. State clearly that the contribution is RLVR training of LLMs to generate programmatic harnesses for bug finding, not the first LLM-based harness, property-based test, or oracle-verifier generation method.

- Add a stronger related-work discussion of property-based testing, parameterized unit tests, generated oracle verifiers, symbolic/concolic testing, coverage-guided test generation, and LLM-assisted fuzzing.

- Add confidence intervals, bootstrap tests, or paired significance tests for GI, ITR, TBR, and test-time scaling results.

- Report matched-budget comparisons. Include token budget, number of model calls, number of generated tests, number of program executions, timeout rates, and wall-clock cost for HarnessLLM, input-output testing, TrickCatcher, and direct prompting.

- Strengthen comparisons to property-based and fuzzing baselines. At minimum, compare against a prompted LLM property-based-testing baseline and a simple randomized/differential-testing baseline. Preferably include established tools or adaptations such as Randoop, EvoSuite, or KLEE where applicable [7,8,9].

- Clarify the reward definition in Equation 1. The current wording around “g fails or f passes, and exists xi: f(xi) != g(xi)” is hard to parse. Spell out how invalid inputs, assertion failures, runtime errors, timeouts, nondeterminism, and output-format mismatches are handled.

- Add a manual false-positive analysis. Sample generated harnesses where the buggy program fails and the reference passes, and verify whether the harness actually tests the intended specification rather than exploiting artifacts of the reference program.

- Expand the human-bug evaluation. The C++ TrickyBugs result is too small to support strong cross-language or human-bug generalization claims, especially because the TBR gain is only 27.1 versus 26.6.

Would strengthen the work:

- Report per-difficulty and per-problem-family results in the main paper, not only aggregate TBR.

- Show more qualitative harness examples, including failures, invalid tests, and cases where invariants are too weak.

- Compare against generating a brute-force reference implementation only, since many successful harnesses seem close to oracle/reference generation.

## References

[1] CodeT: Code Generation with Generated Tests. Bei Chen et al. 2022.

[2] Learning to Generate Unit Tests for Automated Debugging. Archiki Prasad et al. 2025.

[3] Can Large Language Models Write Good Property-Based Tests? Vasudev Vikram et al. 2023.

[4] ALGO: Synthesizing Algorithmic Programs with LLM-Generated Oracle Verifiers. Kexun Zhang et al. 2023.

[5] QuickCheck: A Lightweight Tool for Random Testing of Haskell Programs. Koen Claessen et al. 2000.

[6] Parameterized Unit Tests. Nikolai Tillmann et al. 2005.

[7] EvoSuite: Automatic Test Suite Generation for Object-Oriented Software. Gordon Fraser et al. 2011.

[8] KLEE: Unassisted and Automatic Generation of High-Coverage Tests for Complex Systems Programs. Cristian Cadar et al. 2008.

[9] Fuzz4All: Universal Fuzzing with Large Language Models. Chunqiu Steven Xia et al. 2024.